

# Physiological traits underlying sodicity tolerance in Jamun (*Syzygium cumini* L. Skeels) cultivars

Anshuman Singh[1,2], Ashwani Kumar[1], Jai Prakash[3] and Daya Shankar Mishra[4]

[1] ICAR-Central Soil Salinity Research Institute, Karnal, Haryana, India
[2] ICAR-Central Institute for Subtropical Horticulture, Lucknow, Uttar Pradesh, India
[3] Division of Fruits and Horticultural Technology, ICAR-Indian Agricultural Research Institute, New Delhi, Delhi, India
[4] ICAR-CIAH Central Horticultural Experiment Station, Panchmahals, Gujarat, India

Corresponding author
Anshuman Singh,
anshumaniari@gmail.com

## ABSTRACT

**Background:** There is a lack of research on how sodicity stress affects tree growth and physiological relations in jamun (*Syzygium cumini* L. Skeels). An understanding of cultivar-specific morpho-physiological changes under sodic conditions might aid in the development of more sodicity-tolerant cultivars through genetic improvement, and help identify cultivars suitable for degraded sodic soils.

**Methods:** We assessed the effects of sodicity stress on tree growth, physiological relations, and ion uptake in four cultivars of jamun including CISH J-37 (J-37), CISH J-42 (J-42), Konkan Bahadoli (KB), and Goma Priyanka (GP).

**Results:** Jamun cultivars exhibited varying degrees of reduction in tree growth, leaf area, and gas exchange properties under sodic conditions. Elevated soil pH caused relatively larger declines in trunk cross sectional area (TCSA; >30%) and canopy volume (CV; >25%) in J-42 and KB. Reductions brought on by sodicity stress in leaf area were rather modest (<10%) across cultivars, suggesting that maintaining leaf area may be a key adaptive trait in jamun to cope with sodic conditions. In addition to displaying a notable increase in water use efficiency (WUE), cultivar J-37 also exhibited largely intact levels of relative chlorophyll and photosynthetic rate ($P_n$) under sodic conditions. Despite a high intrinsic $P_n$ under control treatment, cultivar GP displayed a large drop in $P_n$ (37.16%) when exposed to sodicity stress. Comparatively greater increases in leaf phenolics in KB and GP seemed to be at the expense of growth and photosynthesis under sodic conditions. While superoxide dismutase (SOD) and catalase (CAT) activities showed notable spikes in both J-37 and GP, proline accumulation increased substantially only in cultivar J-37 in response to sodicity stress. Despite significant increases in leaf $Na^+$ and $Cl^-$ concentrations, J-37 was found to be comparatively efficient in $Na^+$ and $Cl^-$ exclusion from leaves when compared to other cultivars. Surprisingly, sodicity stress did not alter leaf $K^+$, $Ca^{2+}$ and $Mg^{2+}$ levels noticeably across cultivars. Correlation analysis suggested that elevated leaf $Cl^-$ likely inhibited tree growth more than leaf $Na^+$. Principal component analysis was reasonably efficient in discerning the shared and divergent responses to sodicity stress of the studied cultivars.

**Conclusions:** Membership function analysis revealed a reasonable resilience to sodicity stress only in cultivar J-37. Maintenance of photosynthesis, reduced uptake

of Na$^+$ and Cl$^-$ ions, increased and synergistic activities of SOD and CAT, and a higher leaf K$^+$/Na$^+$ ratio likely accounted for better performance of J-37 trees in sodic soils. Further investigations are needed to elucidate the molecular underpinnings of sodicity tolerance.

## INTRODUCTION

Saline and sodic soils formed due to excessive accumulation of soluble salts and exchangeable sodium, respectively, pose a significant threat to global food security (*Hassani, Azapagic & Shokri, 2020*). Available evidence suggests that over 50% of the global irrigated area may become either saline or sodic by 2050 (*Singh, 2021*). Sodic soils interspersed with less sodic or normal soils occur extensively in semi-arid north-western India. Alkali hydrolysis, formation of sodium carbonates, development of calcic horizon, and degradation of clay minerals under high pH conditions are some major processes responsible for their formation (*Minhas, Yadav & Sharma, 2021*). Sodic soils have low pore spaces, an excess of exchangeable sodium, and poor physical characteristics. High exchangeable Na$^+$ deflocculates such soils, restricting water and air movements. Sub-surface sodicity, especially below 30 cm depth, creates osmotic and ionic stresses, and causes nutritional imbalances (*Sharma & Singh, 2019*). Declining availability of fresh water and amendments, and climate change impacts can hinder the reclamation of sodic soils (*Mandal, 2024*; *Sharma et al., 2016*). While some agroforestry systems and halophytic grasses have potential for remediating sodic soils, they are frequently less remunerative (*Mathur & Mathur, 2024*). Conversely, profitable crops and varieties that can endure excess Na$^+$ are considered a more efficient strategy to improve the productivity of sodic soils (*Melino & Tester, 2023*). However, development of sodicity-tolerant cultivars has progressed rather slowly due to genetic and physiological complexity of sodicity tolerance (*Genc et al., 2019*). Breeding for sodicity tolerance is likely be more successful if selection is based on physiological traits (*Dowla et al., 2021*; *Genc et al., 2019*).

Jamun (*Syzygium cumini* L. Skeels) is widely found throughout the Indian subcontinent, Southeast Asia, and East Africa (*Srivastava & Chandra, 2013*). Jamun fruits are an excellent source of vitamins, minerals, dietary fibre, and bioactive compounds (*Chhikara et al., 2018*; *Madani et al., 2021*). It is grown widely as a multipurpose tree species in traditional agroforestry systems for its fruits, leaves, seeds, and wood (*Sarvade et al., 2016*). Jamun is considered a potential tree species for enhancing food production from wastelands on account of comparatively better tolerance to abiotic stresses. Replanting salt- and waterlogging-affected soils with jamun can also contribute significantly to biodiversity conservation, carbon sequestration, and ecological restoration (*Madani et al., 2021*; *Sarvade et al., 2016*). The carbon sequestration potential of jamun is assessed to be ~63.0 t/ha (*Aruna, 2020*). Growing improved jamun cultivars on sodic lands can enhance

returns to the growers, while simultaneously improving soil quality and halting further degradation (*Datta et al., 2015*).

Feasible strategies including salt-tolerant cultivars are absolutely essential for enhancing fruit production from degraded lands (*Sharma & Singh, 2021*). Although development of high-yielding cultivars has given fillip to the commercial orcharding of jamun (*Singh et al., 2019*), salinity and sodicity problems are still widespread (*Mandal, 2024*) in certain Indian states with substantial potential for jamun cultivation. Because even modest increases in salt stress can adversely affect tree growth and fruit yield even in relatively salt-tolerant fruit crops such as date palm (*Hazzouri et al., 2020*), further improvements in salt tolerance are inevitable, given the progressively worsening soil salinization. Recent studies contradict the widely held notion that jamun is a salt-tolerant species; most genotypes of jamun (40 out of 48) exhibited moderate tolerance to salinity even when salinity of irrigation water was increased gradually (*Singh et al., 2024*). Similarly, jamun has been found to be only moderately tolerant to sodicity stress, with an ESP (exchangeable sodium percentage) threshold of 30.0–40.0% (*Saroj & Sharma, 2017*). Obviously, further improvements in salt tolerance are necessary for sustaining jamun cultivation in salt-affected soils (*Rasheed et al., 2024*).

Salt-induced oxidative, osmotic and ionic stresses hamper the plant growth (*Mahouachi, 2018*) by altering the leaf pigments and gas exchange properties (*Singh et al., 2018a*), impairing the plant water relations (*Larbi et al., 2020*), and disrupting ion homeostasis (*Zahedi et al., 2021*). Plants typically negate such deleterious effects by excluding $Na^+$ and $Cl^-$ ions, maintaining leaf $K^+$ levels, accumulating the compatible osmolytes like proline, and upregulating the anti-oxidant enzymes (*El Moukhtari et al., 2020*; *Sarker & Oba, 2019*; *Singh et al., 2024*; *Zhou-Tsang et al., 2021*). Contrasting genetic variation exists for $Na^+$ and $Cl^-$ uptake (*Hussain et al., 2012*; *Kchaou et al., 2010*), and for osmotic adjustment and anti-oxidant enzymes that scavenge the reactive oxygen species (ROS) (*Regni et al., 2019*). Genetic improvement for sodicity tolerance requires a detailed understanding of such traits and mechanisms (*Dowla et al., 2021*). While a few recent studies have examined the morpho-physiological responses to salinity stress (*Singh et al., 2024*; *Yousaf et al., 2020*), the growth and physiological traits associated with sodicity tolerance still remain enigmatic in jamun. This is primarily because earlier studies assessed the suitability of locally adapted, non-descript jamun genotypes in comparison to other fruit and forestry trees for sodic soils solely based on plant growth, without considering sodicity-induced physiological changes (*Datta et al., 2015*; *Mishra et al., 2014*). The fact that adding amendments to experimental soils can significantly alter plant responses to sodicity stress was also disregarded (*Alcívar et al., 2018*). Because soil $pH_2$ values reported in these studies are often one unit higher than $pH_s$, they might not accurately represent the sodicity that plants actually experience. Their findings also seem to irrelevant to improved cultivars; locally-adapted genotypes usually perform better under salt stress (*Massaretto et al., 2018*).

The growing interest in jamun as a promising fruit crop has necessitated a greater focus on cultivar development programs. Integrating physiological markers with morphological traits can greatly contribute to crop improvement efforts, especially for developing abiotic

stress tolerant and high yielding cultivars for marginal lands. In light of these gaps, we tested four jamun cultivars in a sodic soil to examine how sodicity stress affects tree growth and physiological relations. Our specific objective was to assess the extent to which sodicity stress inhibited vegetative growth, and to identify the physiological responses unique to particular cultivars under sodic conditions.

## MATERIALS AND METHODS

### Study site, experimental material and growth conditions

The experiment was carried out between 2018 and 2020 at Indian Council of Agricultural Research-Central Soil Salinity Research Institute located in Karnal, Haryana, India. The study area has a semi-arid climate with ~700 mm of annual rainfall. The experimental farm soils still suffer from moderate to high sodicity in the sub-surface soil. One-year old plants of jamun cultivars CISH J-37 (J-37), CISH J-42 (J-42), Goma Priyanka (GP), and Konkan Bahadoli (KB), obtained from ICAR-CHES, Godhra, Gujarat, were transferred to the experimental field in September, 2018 following acclimation. Soil analysis revealed considerable spatial and vertical differences in soil sodicity in terms of soil $pH_s$ and exchangeable sodium percentage (ESP) (*Hassani, Azapagic & Shokri, 2020*). Soil $pH_s$ was 7.81, 8.14, and 8.35 (average 8.09) in control and 8.45, 8.82, and 9.11 (8.80) in sodic treatment at 0–30, 30–60, and 60–100 cm depths, respectively. The corresponding ESP was 8.35%, 10.82%, and 16.58%, respectively, in control and 18.34%, 29.79%, and 39.60%, respectively, in sodic treatment. The scion shoots of all the cultivars were grafted on 1-year-old uniform rootstock plants raised by sowing the seeds from a single mother tree (~30 y) in polybags (25 cm × 10 cm) filled with soil and farm yard manure (3:1). Rootstock seedlings uniform in growth and having pencil thick stems were used in grafting using ~20 cm long healthy scion shoots (*Singh et al., 2011*). The plants were transplanted into auger-holes (diameter: 30 cm, depth: 120 cm) filled with the original soil without the addition of any soil amendment. A square system of planting was used, keeping between- and within-row distances of 6 m each. The young plants were trained properly to create a strong canopy framework. This was achieved by retaining on 3–5 well-spaced scaffold branches in different directions of the main stem above 60 cm from the ground surface. These branches served as the main framework for canopy development. Subsequent to planting, each tree was fertilized using 125 g N, 50 g $P_2O_5$ and 50 g $K_2O$ in two equal splits. While half the dose was applied immediately after transplanting, the remaining half was applied in the ensuing June month (*Singh et al., 2010*). The irrigation water was applied into 1 m wide irrigation channels at fortnightly intervals during autumn-winters (October-February) and at weekly intervals during summers (March-June). The irrigation water had an electrical conductivity of 0.68 dS/m, pH of 8.04, $Na^+$ of 2.43, $K^+$ of 0.16, $Ca^{2+} + Mg^{2+}$ of 4.14, $Cl^-$ of 0.89 and $HCO^-_3$ of 4.22 (me/l). The irrigation water was safe for irrigation, and unlikely to induce soil sodicity even after long-term use (*Choudhary, 2017*).

### Soil properties

After about a year of planting, there were notable differences in tree growth, likely due to innate variations in soil properties (*Marino et al., 2019*; *Peterson, Helgason & Ireson, 2019*).

Subsequently, twelve trees of each jamun cultivar were randomly selected for recording the spatial variations in soil pH, electrical conductivity, ESP, cation exchange capacity (CEC), and organic carbon (OC). Soil samples were collected using an auger from 0–30, 30–60, and 60–100 cm depths, approximately 50 cm from 2–3 directions of the trunk. The soils from the same depths were mixed to prepare the composite representative samples. Then, soil samples were air dried and sieved (2.0 mm sieve). Soil pH ($pH_s$) and saturation extract electrical conductivity ($EC_e$) were determined using digital pH and conductivity metres (Eutech, Singapore). The CEC and ESP were measured using the procedures described in Bhargava (2003). For CEC determination, the soils samples (5 g) were treated with 33 ml of 1N sodium acetate (pH 8.2) for replacing the exchangeable cations by $Na^+$ ions. Ethanol was used to remove any excess sodium acetate. The adsorbed sodium was then replaced from the sample by extraction with three portions of 1N ammonium acetate. Following dilution to 100 ml, Na concentration was measured using a flame photometer (Systronics, India). The CEC was calculated using the formula:

$$CEC\ (me/100\ g) = \frac{Na\ concentration\ (me/l)\ x\ 10}{soil\ sample\ (g)}$$

where CEC is the cation exchange capacity (milliequivalent per 100 g of soil), Na concentration refers to the $Na^+$ concentration in milliequivalent per litre (me/l) from the standard curve, and soil sample (g) refers to the weight of soil sample (5 g) used.

For ESP measurement, 33 ml of 60% ethanol was poured gradually and allowed to leach. This procedure was repeated until the leachate's electrical conductivity dropped below 40 micromhos/cm. The leachate was discarded. Following addition of three portions of 1N ammonium acetate (33 ml each) in a 100 ml volumetric flask, Na concentration was determined. The exchangeable sodium (me/100 g of soil) was calculated by the formula:

$$ES\ (me/100\ g) = \frac{Na\ concentration\ (me/l)\ x\ 10}{soil\ sample\ (g)}$$

where, ES is the exchangeable sodium (milliequivalent per 100 g of soil), Na concentration refers to the $Na^+$ concentration of extract in milliequivalent per litre from the standard curve, and soil sample (g) refers to the weight of soil sample (5 g) used.

Exchangeable sodium percentage (ESP; %) was computed using the formula:

$$ESP\ (\%) = \frac{ES\ x\ 100}{CEC}$$

where ESP (%) is the exchangeable sodium percentage, ES refers to the exchangeable sodium (milliequivalent per 100 g of soil), and CEC stands for cation exchange capacity (milliequivalent per 100 g of soil).

## Tree growth and leaf area

Six trees ($n = 6$) of each jamun cultivar within both control and sodicity treatments were tagged for recording the observations. Tree height, canopy spread and trunk diameter were recorded during the first week of September, 2020; after 2 years after planting. Trunk diameter was measured in the east-west (E-W) and north-south (N-S) directions 20 cm

above the graft union using a digital caliper (Mitutoyo, Kanagawa, Japan). Trunk cross sectional area (TCSA) was calculated using the formula: TCSA = $\pi(d/2)^2$; where d = mean of E-W and N-S trunk diameters. Canopy volume (CV) was calculated as: CV = $(w^2 \times h)/2$; where w= canopy diameter in E-W and N-S directions, and h= tree height. Four fully developed leaves in the outer canopy of each replicate tree were tagged for measuring the unit leaf area using a handheld laser leaf area meter (CI-203; CID Bio-Science). There values were averaged and treated as a single replication in data analysis.

## Leaf phenolics, relative chlorophyll and gas exchange traits

The leaf phenolics, including anthocyanins (anth) and flavonols (flav), relative chlorophyll, and gas exchange attributes, were measured on the same leaves that had previously been used for leaf area measurement. The leaf phenolics were measured using the Dualex 4 Scientific leaf-clip sensor (FORCE-A, Orsay Cedex, France). This leaf-clip sensor allows rapid and non-destructive measurements of leaf flavonols and anthocyanins. It first measures near-infrared chlorophyll fluorescence under a first reference excitation light not absorbed by polyphenols. It then measures the specific light absorbed by polyphenols including the green light for anthocyanins and ultraviolet light for flavonols. The dual-light excitation strategy enables quantification of polyphenols without damaging the leaves, and helps understand the changes in their levels in response to stress conditions (*ForceA, 2019*). A Soil Plant Analysis Development (SPAD) meter (SPAD-502Plus) was used to determine the relative leaf chlorophyll content. A portable infrared gas analyzer attached to a 6 cm$^2$ cuvette (LI-COR 6400 XT system; LI-COR, Lincoln, NE, USA) was used to assess the gas exchange attributes: net photosynthesis ($P_n$), transpiration ($E$), stomatal conductance ($g_s$), and internal $CO_2$ concentration ($C_i$). Before being enclosed in the leaf chamber, the leaves were wiped-off with a muslin cloth. A photosynthetic photon flux density of 1,000 $\mu$mol m$^{-2}$ s$^{-1}$, $CO_2$ concentration of 400 ppm, and a leaf temperature of 25 °C of were maintained during the measurements. The ratio of net photosynthetic to transpiration rate ($P_n/E$) was used to compute the instantaneous water use efficiency (WUE). These measurements were performed on 7 and 8 September, 2020 between 9.0 and 11.0 h.

## Anti-oxidant enzymes and proline

Fresh leaf sample (300 mg) was homogenized in 0.1 M phosphate buffer (pH 7.5) supplemented with 5% (w/v) polyvinyl polypyrrolidone, 1 mM EDTA, and 10 mM b-mercapto-ethanol to assess ascorbate peroxidase (APX, EC 1.11.1.11) and superoxide dismutase (SOD, EC 1.15.1.11) activities. APX activity was measured subsequent to the oxidation of ascorbic acid (*Nakano & Asada, 1981*). The reaction mixture had 2.25 ml of 100 mM phosphate buffer (pH 7.0), 0.2 ml of 0.5 mM ascorbate, 0.2 ml of 0.1 mM $H_2O_2$, and 0.05 ml of the enzyme extract. Measurements using a UV-VIS spectrophotometer (Specord 210 Plus; Analytik Jena, Jena, Germany) revealed a drop in absorbance at 290 nm, indicating the oxidation of ascorbic acid. The enzyme activity was determined using ascorbic acid's molar extinction coefficient of 2.8 mM$^{-1}$ cm$^{-1}$. One enzyme unit refers to one $\mu$ mole of ascorbic acid oxidized per minute at 290 nm. The ability of SOD to

prevent nitro blue tetrazolium (NBT) from being reduced photochemically was used to measure its activity (*Beauchamp & Fridovich, 1971*). The reaction mixture (3.0 ml) consisted of 2.5 ml of 50 mM Tris-HCl (pH 7.8), and 100 µl each of 14 mM L-methionine, 10 µM NBT, 3 µM riboflavin, 0.1 mM EDTA, and the enzyme extract. The tubes were carefully shaken and positioned 30 cm below three 20 W fluorescent light bulbs (Phillips, India). Reaction was stopped after 40 min of incubation. The tubes were covered with black cloth to shield them from light. A non-irradiated colourless reaction mixture was used as control. The reaction mixture without enzyme extract developed a bright color and showed a decrease in absorbance following the addition of enzyme (Specord 210 Plus; Analytik Jena, Jena, Germany). The amount of enzyme needed to prevent one mmol of NBT from being photo-reduced was determined to be one unit of SOD. Catalase (CAT; EC 1.11.1.6) activity was determined using the procedure given in *Aebi (1984)*. Reaction mixture comprised 0.5 ml of 0.1 M phosphate buffer (pH 7.0), 0.4 ml of 0.2 M hydrogen peroxide, and 0.1 ml of the diluted enzyme extract. The reaction was stopped by adding 3 ml mixture of glacial acetic acid (1:3 v/v) and potassium dichromate (5% w/v) after incubation at 37 °C for 3 min. The tubes were heated in a bath of boiling water for 10 min. A control was run under similar conditions. The absorbance of the test and control tubes was measured at 570 nm following their cooling (Specord 210 Plus). The absorbance of the test samples was subtracted from the absorbance of the control to calculate the amount of residual $H_2O_2$. One unit of enzyme activity refers to the enzyme quantity required to catalyze the oxidation of 1.0 µmole $H_2O_2$ per minute. The rate of guaiacol oxidation in the presence of $H_2O_2$ at 470 nm was used to measure the activity of peroxidase (POX, EC 1.11.1.7) (*Rao, Paliyath & Ormrod, 1996*). The reaction mixture consisted of 50 mM phosphate buffer (pH 6.5; 2.5 ml), 0.5% hydrogen peroxide (0.1 ml), 0.2% 0-dianisidine (0.1 ml), and the enzyme extract (0.1 ml). $H_2O_2$ (0.1 ml) was added to initiate the reaction. The reaction mixture devoid of $H_2O_2$ served as the blank. The change in absorbance was monitored at 430 nm for 3 min. One unit of peroxidase is represented by a change of one optical density in a minute. For proline extraction, the leaf tissue (200 mg) was homogenized in 10 ml of 3% sulphosalicyclic acid (*Bates, Waldren & Teare, 1973*). Acid-ninhydrin and glacial acetic acid (2 ml) were added to the extract (2 ml). The reaction was stopped in a water bath after 1 h of incubation at 100 °C. Subsequent to extraction using 4 ml of toluene, reaction mixture was rapidly stirred for 15–20 s. The absorbance was measured at 520 nm using toluene as blank (Specord 210 Plus, Germany).

## Leaf ions

Leaf samples, dried to a constant weight at 60 °C in a hot air oven (NSW, India), were finely ground and 100 mg tissue sample was digested in nitric acid ($HNO_3$). $Na^+$ and $K^+$ were determined using a flame photometer (Systronics India), $Cl^-$ using a chloride ion-selective electrode (Thermo Fisher Scientific, Mumbai, India), and $Ca^{2+}$ and $Mg^{2+}$ using an Atomic Absorption Spectrophotometer (Analytik Jena, Jena, Germany). All ion contents are in mg/g dry weight.

## Statistical analysis

The experiment was laid out in a randomized block design with six replications ($n = 6$) within both control and sodicity stress treatments. Each replication consisted of a single biological replicate. The growth observations were recorded on these six trees ($n = 6$) of each cultivar under both control and sodic treatments. Leaf area, leaf phenolics, relative chlorophyll (SPAD) and gas exchange attributes were measured on four leaves of each replicate tree; their values were averaged and treated as a single replication in data analysis ($n = 4$). A two-way analysis of variance (ANOVA) was used to examine the independent and interaction effects of cultivar and sodicity on different traits. The means were compared by Tukey test ($p < 0.05$). Pearson's correlations between the measured traits and the corresponding significance levels were computed. These analyses were performed using the MV App (*Julkowska et al., 2019*). Principal component analysis (PCA) was applied on replicated observations to discern the major patterns in the data (JAMOVI 2.3.28).

## Membership function analysis

Membership function analysis, a comprehensive index based on multiple traits was used to rank the jamun cultivars for sodicity tolerance. This approach is particularly useful for assessing traits such as salt tolerance which are influenced by multiple traits and cannot be reliably measured using a single variable (*Tian et al., 2024*). This methodology uses membership functions based on the theory of fuzzy mathematics. Membership functions are used to define fuzzy sets like 'salt stress'. Fuzzy logic accepts partial membership, assigning values between 0 and 1, in contrast to classical logic, which stipulates that an element is either a member (1) or not a member (0) of a set (*Ross, 2004*). The membership function values (MFV) for each measured trait were calculated using the sodicity tolerance coefficients (STCs). The STCs were calculated as the ratio of a trait's value under sodicity stress to its value under control (non-sodic) conditions, expressed as percentage. Next, the MFV1 for each trait was calculated (*Quamruzzaman et al., 2022*). The traits inversely related with sodicity tolerance (leaf $Na^+$ and $Cl^-$) were measured using MFV2. Leaf $Na^+$ and $Cl^-$ are considered to have inverse relationships with sodicity tolerance; their exclusion and/or compartmentalization and thus their lower concentrations in leaves would improve tolerance to sodicity stress. To calculate MFV2, the MFV1 values for leaf $Na^+$ and $Cl^-$ were subtracted from 1. Each trait has its own MFV, ranging between '0' and '1'. Higher salt tolerance is correlated with higher mean MFV, and *vice versa*. The MFVs of each measured trait were averaged to determine the mean MFV for jamun cultivars (*Gyanagoudar et al., 2024*). The sodicity tolerance levels of jamun cultivars were divided into five classes based on the mean value ($\bar{X}$) and standard deviation (SD) of the mean MFV ($X_i$) at 90% and 68% confidence intervals (Z score = 1.64 and 1, respectively). These five classes included 'highly tolerant': $X_i \geq \bar{X} + 1.64 \times SD$; 'tolerant': $\bar{X} + 1.64 \times SD > X_i \geq \bar{X} + 1 \times SD$; 'moderately tolerant': $\bar{X} + 1 \times SD > X_i > \bar{X} - 1 \times SD$; 'sensitive': $\bar{X} - 1 \times SD > X_i > \bar{X} - 1.64 \times SD$; and 'highly sensitive': $\bar{X} - 1.64 \times SD > X_i$ (*Gyanagoudar et al., 2024*).

## RESULTS

### Soil properties

The soil $pH_s$ and ESP increased with depth, and were invariably significantly higher at different depths in sodic than in normal soils. The $EC_e$ values were 0.58, 0.73 and 0.92 dS/m in control, and 0.66, 0.95 and 1.14 dS/m in sodic treatment in 0–30, 30–60, and 60–100 cm depths. The corresponding ESP was 8.35, 10.82, and 16.58%, respectively, in control and 18.34%, 29.79%, and 39.60%, respectively, in sodic treatment. The CEC was significantly lower in sodic than in control treatment across different depths. The organic carbon (OC) declined significantly with increase in sodicity across all the soil depths (Table S1). Pearson's correlation analysis indicated significant positive relationships of CEC ($r = 0.946$) and ESP ($r = 0.918$) with the respective concentrations of $Na^+$ measured during these determinations. Soil ESP had significant positive correlations with pH ($r = 0.943$), and negative correlations with CEC ($r = -0.871$) and OC ($r = -0.914$) (Table S2).

### Tree growth

The effects of cultivar (C), sodicity (S), and their interaction (C × S) were significant ($p < 0.05$) on trunk cross sectional area (TCSA), canopy volume (CV), and leaf area (LA). Cultivars GP and KB had the highest (18.31 $cm^2$), and the lowest (10.49 $cm^2$) TCSA under control conditions. When compared to controls, sodicity stress reduced TCSA by 18.03, 31.29, 21.68, and 32.60% in J-37, J-42, GP, and KB, respectively. Under control conditions, CV was the largest (2.37 $m^3$) in GP, and the smallest (0.77 $m^3$) in J-42. Sodicity-induced reductions in CV were significant in cultivars J-42 (25.97%), GP (22.79%), and KB (32.60%). Sodicity stress caused significant reductions in LA only in cultivars J-37 (6.22%), GP (5.36%), and KB (8.19%) (Table 1).

### Relative chlorophyll and gas exchange attributes

Compared to controls, sodicity-induced reductions in SPAD were relatively large (~17.0%) in J-42 and GP. Sodicity-induced declines in $P_n$ were fairly similar in cultivars J-37 and J-42 (~18.0%). Cultivar GP exhibited the largest drop (37.16%) in $P_n$ under sodicity stress. The tested cultivars displayed varying reductions in $E$ [J-37 (28.42%), J-42 (21.34%), GP (37.23%), and KB (35.58%)] in response to sodicity. The $g_s$ also declined in a similar fashion; decreases relative to controls were 30.77, 20.0, 47.06, and 42.86% in J-37, J-42, GP, and KB, respectively. Interestingly, there were indiscernible differences among the cultivars [J-37 (15.80%), J-42 (19.97%), GP (19.21%), and KB (16.31%)] for sodicity-induced reductions in $C_i$. While sodicity stress did not significantly affect WUE in J-42 and GP, cultivars J-37 (13.49%) and KB (21.91%) exhibited significant increases in WUE (Table 2).

### Leaf flavonols, anti-oxidant enzymes and proline

Sodicity-induced upticks in anthocyanins (Anth) differed substantially among the cultivars; KB displayed the biggest increase (80.0%) while J-42 showed the lowest uptick

**Table 1 Effects of sodicity stress on vegetative growth in jamun cultivars.**

| Cultivar | Treatment | TCSA | CV | LA |
|---|---|---|---|---|
| CISH J-37 | Control | 11.37 ± 0.32c | 0.89 ± 0.06cd | 89.76 ± 2.13a |
| | Sodic | 9.32 ± 0.51d | 0.75 ± 0.06de | 84.18 ± 1.49bc |
| CISH J-42 | Control | 14.03 ± 0.76b | 0.77 ± 0.04d | 84.28 ± 1.34bc |
| | Sodic | 9.64 ± 0.39d | 0.57 ± 0.04e | 82.20 ± 1.32cd |
| Goma Priyanka | Control | 18.31 ± 1.29a | 2.37 ± 0.28a | 86.75 ± 1.39ab |
| | Sodic | 14.34 ± 1.32b | 1.83 ± 0.08b | 82.10 ± 2.03cd |
| Konkan Bahadoli | Control | 10.49 ± 0.65c | 1.04 ± 0.11c | 80.38 ± 2.08d |
| | Sodic | 7.07 ± 0.31e | 0.75 ± 0.08de | 73.80 ± 1.71e |
| F-value | | | | |
| Cultivar (C) | | *** | *** | *** |
| Sodicity (S) | | *** | *** | *** |
| C x S | | ** | ** | * |

Notes:
TCSA- trunk cross sectional area ($cm^2$), CV- canopy volume ($m^3$), LA- leaf area ($cm^2$), Each value represents mean ± SD. Mean with a common letter within each column are not statistically different ($p$ 0.05).
*** $p < 0.001$.
** $p < 0.01$.
* $p < 0.05$.

**Table 2 Effects of sodicity stress on leaf relative chlorophyll and gas exchange attributes in jamun cultivars.**

| Genotype | Treatment | SPAD | $P_n$ | E | $g_s$ | $C_i$ | WUE |
|---|---|---|---|---|---|---|---|
| CISH J-37 | Control | 47.42 ± 1.27b | 8.65 ± 0.23b | 2.85 ± 0.11d | 0.13 ± 0.02cd | 283.29 ± 4.62b | 3.04 ± 0.13b |
| | Sodic | 45.05 ± 1.79b | 7.03 ± 0.16cd | 2.04 ± 0.08g | 0.09 ± 0.01e | 238.54 ± 3.43e | 3.45 ± 0.07a |
| CISH J-42 | Control | 40.42 ± 1.44c | 9.16 ± 0.21b | 3.28 ± 0.19c | 0.15 ± 0.02bc | 253.65 ± 6.18d | 2.80 ± 0.13bc |
| | Sodic | 33.97 ± 1.08d | 7.55 ± 0.32c | 2.58 ± 0.12ef | 0.12 ± 0.01de | 203.01 ± 3.41f | 2.93 ± 0.15b |
| Goma Priyanka | Control | 41.60 ± 0.97c | 10.63 ± 0.59a | 3.76 ± 0.12b | 0.17 ± 0.02b | 290.68 ± 3.92b | 2.83 ± 0.19bc |
| | Sodic | 34.55 ± 1.72d | 6.68 ± 0.44d | 2.36 ± 0.07f | 0.09 ± 0.01e | 234.83 ± 3.34e | 2.83 ± 0.25bc |
| Konkan Bahadoli | Control | 51.20 ± 1.19a | 9.02 ± 0.25b | 4.30 ± 0.10a | 0.21 ± 0.03a | 313.62 ± 6.38a | 2.10 ± 0.03d |
| | Sodic | 46.37 ± 1.76b | 7.04 ± 0.14cd | 2.77 ± 0.20de | 0.12 ± 0.03de | 263.03 ± 4.25c | 2.56 ± 0.21c |
| F-value | | | | | | | |
| Cultivar (C) | | *** | *** | *** | *** | *** | *** |
| Sodicity (S) | | *** | *** | *** | *** | *** | *** |
| C x S | | ** | *** | *** | *** | * | ** |

Notes:
SPAD-relative leaf chlorophyll, $P_n$: net photosynthesis (µmol/m/s), $E$: transpiration rate (mmol/m/s), $g_s$: stomatal conductance (mol/m/s), $C_i$: internal $CO_2$ concentration (µmol/mol), WUE: water use efficiency (µmol $CO_2$ mmol $H_2O^{-1}$). Each value represents mean ± SD. Means with a common letter within each column are not statistically different ($p$ 0.05).
*** $p < 0.001$.
** $p < 0.01$.
* $p < 0.05$.

(52.18%) in comparison to controls. Except for KB (0.55 units), the flavonol (Flav) levels in other tested cultivars were statistically similar (0.63–0.67 units) in control. While both J-37 (31.75%) and J-42 (29.23%) showed relatively less increases in Flav, GP (40.30%) and KB (70.91%) exhibited moderate and high increases in Flav, respectively, under sodic conditions (Table 3). While the constitutive levels of APX were hardly different, sodicity

**Table 3 Effects of sodicity stress on leaf flavonols, anti-oxidant enzymes and proline in jamun cultivars.**

| Cultivar | Treatment | Anth | Flav | APX | CAT | POX | SOD | Proline |
|---|---|---|---|---|---|---|---|---|
| CISH J-37 | Control | 0.19 ± 0.02cd | 0.63 ± 0.02c | 2.93 ± 0.09e | 1.72 ± 0.13g | 6.88 ± 0.12d | 4.19 ± 0.15f | 1.28 ± 0.06f |
| | Sodic | 0.34 ± 0.03ab | 0.83 ± 0.04b | 4.31 ± 0.14d | 3.35 ± 0.12bc | 7.32 ± 0.14c | 8.08 ± 0.14a | 3.87 ± 0.07b |
| CISH J-42 | Control | 0.23 ± 0.02c | 0.65 ± 0.03c | 4.24 ± 0.08d | 2.87 ± 0.10e | 5.16 ± 0.10e | 6.61 ± 0.12b | 0.97 ± 0.06g |
| | Sodic | 0.35 ± 0.03ab | 0.84 ± 0.03b | 4.60 ± 0.15c | 3.09 ± 0.09d | 7.14 ± 0.08c | 6.75 ± 0.10b | 1.42 ± 0.10f |
| Goma Priyanka | Control | 0.18 ± 0.02d | 0.67 ± 0.03c | 3.16 ± 0.11e | 2.27 ± 0.10f | 4.85 ± 0.14f | 2.43 ± 0.23g | 1.83 ± 0.07e |
| | Sodic | 0.31 ± 0.02b | 0.94 ± 0.04a | 6.52 ± 0.19a | 4.74 ± 0.13a | 7.91 ± 0.09b | 5.34 ± 0.10d | 4.07 ± 0.14a |
| Konkan Bahadoli | Control | 0.20 ± 0.02cd | 0.55 ± 0.03d | 3.08 ± 0.12e | 3.25 ± 0.09cd | 8.06 ± 0.12ab | 4.65 ± 0.14e | 2.11 ± 0.10d |
| | Sodic | 0.36 ± 0.03a | 0.94 ± 0.04a | 5.18 ± 0.11b | 3.46 ± 0.11b | 8.15 ± 0.09a | 6.04 ± 0.14c | 3.18 ± 0.10c |
| F-value | | | | | | | | |
| Cultivar (C) | | *** | *** | *** | *** | *** | *** | *** |
| Sodicity (S) | | *** | *** | *** | *** | *** | *** | *** |
| C x S | | ns | *** | *** | *** | *** | *** | *** |

**Notes:**
Anth, anthocyanins (units); Flav-flavonols (units), APX, ascorbate peroxidase; CAT, catalase; POX, peroxidase; SOD, superoxide dismutase (all antioxidant enzymes in units/g fresh weight); proline (mg/g fresh weight). Each value represents mean ± SD. Means with a common letter within each column are not statistically different ($p$ 0.05).
*** $p < 0.001$, ns: non-significant.

stress increased the APX activity by 47.09, 8.49, 106.33, and 68.18% in J-37, J-42, GP, and KB, respectively, relative to controls. The CAT levels varied considerably among cultivars under both control and sodicity treatments; GP (108.81%) and KB (6.46%) exhibited the highest and lowest spikes in CAT activity under sodic conditions (Table 3). These cultivars also showed the highest (63.09%) and the lowest (1.12%) increases in POX activity in response to sodicity stress. Sodicity-induced increases in SOD activity were remarkable only in J-37 (92.84%) and GP (119.75%), and rather weak in J-42 (2.12%). The tested cultivars also differed remarkably in leaf proline levels under sodic conditions; increases in proline were substantially greater in J-37 (202.34%) and GP (122.40%) than in J-42 (46.39%) and KB (50.71%) (Table 3).

## Leaf ions

Leaf $Na^+$ increased markedly in response to sodicity stress; cultivars KB (170.99%) and J-37 (86.36%) displayed the largest and smallest upticks in leaf $Na^+$. Although leaf $K^+$ dropped under sodicity stress, declines were rather subtle: 5.66% in J-37, 7.04% in J-42, 16.17% in GP, and 1.37% in KB. While J-37 exhibited modest increase (11.96%) in leaf $Ca^{2+}$ when exposed to sodicity stress, leaf $Ca^{2+}$ levels did not differ significantly between control and sodicity treatments in J-42, KB and GP (Table 4). Under sodicity stress, leaf $Mg^{2+}$ decreased significantly in J-37 (9.75%), J-42 (27.11%), and KB (9.69%). Despite having the highest leaf $Cl^-$ under control (1.91 mg/g DW), cultivar J-37 showed the lowest increase (33.51%) in leaf $Cl^-$ under sodic conditions. In contrast, J-42 (131.82%), GP (172.09%), and KB (121.01%) showed noticeably larger increases in leaf $Cl^-$ in response to sodicity stress. Sodicity stress caused significant reductions in leaf $K^+/Na^+$ ratio; varying between 49.55% (J-37) and 63.68% (KB) (Table 4).

**Table 4 Effects of sodicity stress on leaf mineral ions and $K^+/Na^+$ ratio in jamun cultivars.**

| Cultivar | Treatment | $Na^+$ | $K^+$ | $Ca^{2+}$ | $Mg^{2+}$ | $Cl^-$ | $K^+/Na^+$ |
|---|---|---|---|---|---|---|---|
| CISH J-37 | Control | 1.98 ± 0.08c | 4.42 ± 0.07a | 3.68 ± 0.10d | 2.36 ± 0.06e | 1.91 ± 0.05d | 2.24 ± 0.09a |
| | Sodic | 3.69 ± 0.19b | 4.17 ± 0.08b | 4.12 ± 0.09b | 2.13 ± 0.07f | 2.55 ± 0.08b | 1.13 ± 0.07d |
| CISH J-42 | Control | 2.06 ± 0.09c | 2.84 ± 0.13de | 3.95 ± 0.09c | 4.02 ± 0.10a | 1.32 ± 0.07e | 1.38 ± 0.07c |
| | Sodic | 4.17 ± 0.08a | 2.64 ± 0.16e | 4.02 ± 0.11bc | 2.93 ± 0.08b | 3.06 ± 0.09a | 0.64 ± 0.05f |
| Goma Priyanka | Control | 1.88 ± 0.10c | 3.34 ± 0.14c | 4.56 ± 0.06a | 2.83 ± 0.07bc | 0.86 ± 0.03f | 1.79 ± 0.12b |
| | Sodic | 3.57 ± 0.11b | 2.80 ± 0.11de | 4.41 ± 0.10a | 2.67 ± 0.08cd | 2.34 ± 0.08c | 0.79 ± 0.04e |
| Konkan Bahadoli | Control | 1.31 ± 0.08d | 2.92 ± 0.10d | 2.26 ± 0.11e | 2.58 ± 0.13d | 1.38 ± 0.06e | 2.23 ± 0.06a |
| | Sodic | 3.55 ± 0.09b | 2.88 ± 0.12d | 2.36 ± 0.07e | 2.33 ± 0.08e | 3.05 ± 0.10a | 0.81 ± 0.05e |
| F-value | | | | | | | |
| Cultivar (C) | | *** | *** | *** | *** | *** | *** |
| Sodicity (S) | | *** | *** | *** | *** | *** | *** |
| C x S | | *** | *** | *** | *** | *** | *** |

Notes:
All ions are in mg/g dry weight. Each value represents mean ± SD. Means with a common letter within each column are not statistically different ($p$ 0.05).
*** $p < 0.001$.

## Correlation analysis

Pearson's bivariate correlations and the corresponding $p$-values are given in Table S2 and Fig. 1. TCSA had highly significant ($p = 0.000$) positive correlations with CV ($r = 0.821$), LA (0.549), $P_n$ (0.625), and $Ca^{2+}$ (0.638), and significant negative correlations with Anth (0.609), POX (0.721), SOD (0.626), and $Cl^-$ (0.747). Similarly, CV had significant positive correlations with $P_n$ (0.414), $E$ (0.292), $C_i$ (0.318), $Ca^{2+}$ (0.445), and $Mg^{2+}$ (0.437), and significant negative correlations with Anth (0.418), POX (0.358), SOD (0.734), and $Cl^-$ (0.547). Presumably, trees with bigger trunk diameters can translocate more water and nutrients for the enhanced growth of leaves and canopies. Significant positive correlations of TCSA and CV with $P_n$ were likely because more leaves and a bigger canopy in trees with larger trunk diameters increase the surface area for photosynthesis. Calcium and magnesium play structural and functional roles in the leaves, and their increased accumulation may support the metabolic demands of a larger canopy. The negative correlations of TCSA and CV with anthocyanins and antioxidant enzymes (POX and SOD) highlight trade-off between tree growth and stress-related secondary metabolism. Interestingly, both TCSA and CV exhibited stronger negative correlations with leaf $Cl^-$ (0.747 and 0.547, respectively) than with leaf $Na^+$ (0.439 and 0.282, respectively), implying that increased leaf $Cl^-$ levels may more adversely impact jamun tree growth under sodic conditions. Leaf area (LA) correlated positively with $P_n$ (0.460), WUE (0.433), $K^+$ (0.582), $Ca^{2+}$ (0.611), and $K^+/Na^+$ ratio (0.482). Larger leaves probably contribute more to photosynthesis because of their larger surface area for light interception, higher density of mesophyll cells, and increased gas exchange capacity. Both $K^+$ and $Ca^{2+}$ play diverse physiological functions in plants, including the maintenance of cell osmotic pressure which may explain their strong positive correlations with leaf area. LA showed significant negative correlations with flavonols, anti-oxidant enzymes, proline, and $Na^+$ and $Cl^-$; its correlations with Anth (0.544), Flav (0.485), CAT (0.547), and POX (0.552) were notably

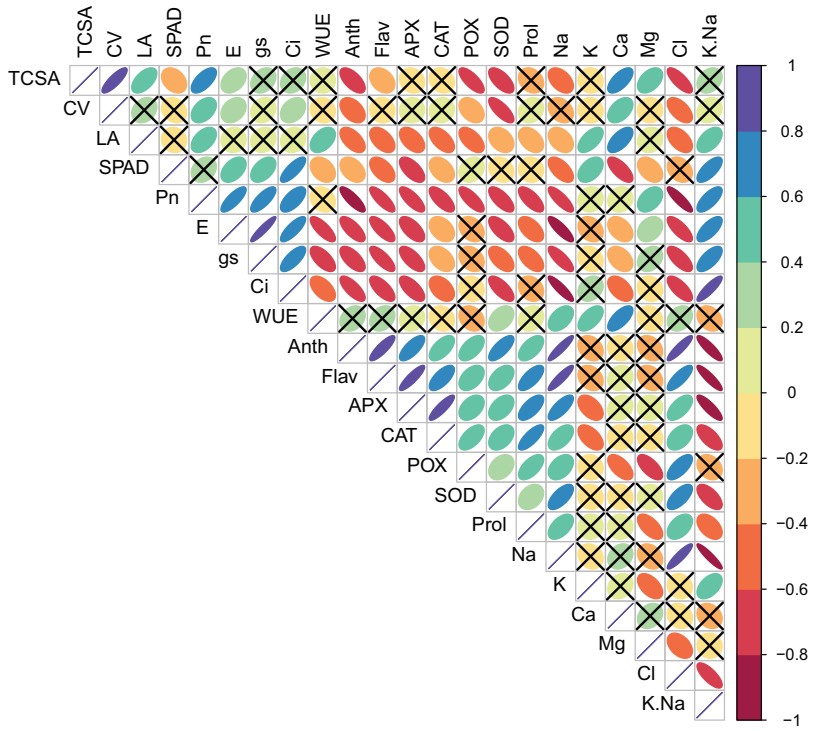

**Figure 1 Correlation plot showing the Pearson's correlation coefficients between the measured traits.** Ellipse color and size indicate the strength of correlation. Non-significant correlations ($p > 0.05$) are shown with a cross in the individual cells. TCSA-Trunk cross sectional area, CV-canopy volume, LA-leaf area, Anth-leaf anthocyanins, Flav-leaf flavonols, SPAD-relative leaf chlorophyll, $P_n$-net photosynthesis, $E$-Transpiration rate, $g_s$-stomatal conductance, $C_i$-internal $CO_2$ concentration, WUE-water use efficiency, APX-ascorbate peroxidase, CAT-catalase, POX-peroxidase, SOD-superoxide dismutase, Prol-proline, Na-leaf $Na^+$, K-leaf $K^+$, Ca-leaf $Ca^{2+}$, Mg-leaf $Mg^{2+}$, Cl-leaf $Cl^-$, K/Na-leaf $K^+/Na^+$ ratio.

negative. Plants under stress often accumulate more secondary metabolites including anthocyanins, flavonols, and antioxidant enzymes at the expense of growth. While gas exchange attributes ($P_n$, $g_s$ and $C_i$) had highly significant positive correlations with each other, WUE exhibited highly significant negative correlations with $E$ (0.751), $g_s$ (0.623), and $C_i$ (0.523) (Table S2, Fig. 1).

## Principal component analysis

The results of Bartlett's test ($\chi^2 = 2{,}293.54$, $p < 0.001$) and a reasonably high Kaiser-Meyer-Olkin score (0.754) suggested that PCA would efficiently reduce the dimensionality, producing distinct principal components. The first four principal components (Eigen value >1.0) accounted for 90.50% of the cumulative variance in data. The PC1 (Eigen value = 10.57, variance = 48.10%) was strongly correlated with gas exchange parameters ($P_n$, $E$, $g_s$ and $C_i$), flavonols (Anth and Flav), APX, $Na^+$ and $Cl^-$. The PC2 (Eigen value = 4.16, variance = 18.90%) was largely a construct of SPAD, POX and $Ca^{2+}$. While PC3 (Eigen value = 3.01, variance = 13.70%) had leaf $K^+$ and WUE as the highly loaded variables, PC4 (Eigen value = 2.17, variance = 9.90%) had CV and leaf proline as the most prominent variables (Table S3). A glance at the PCA biplot indicated that the grouping of

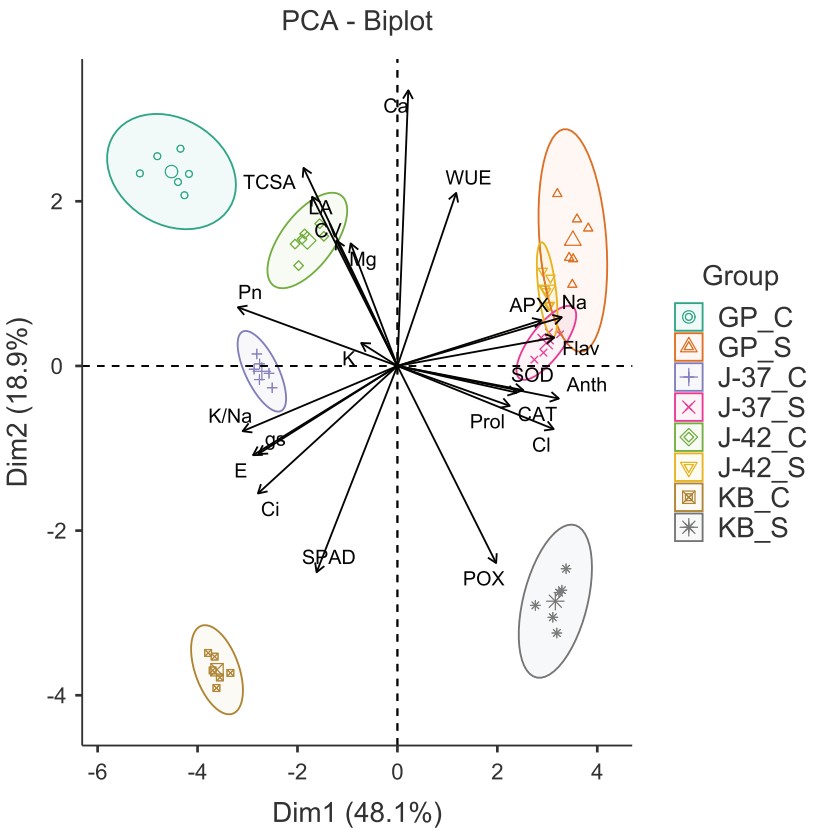

**Figure 2 Principal component analysis biplot displaying the loading of traits and groups.** TCSA-Trunk cross sectional area, CV-canopy volume, Anth-leaf anthocyanins, Flav-leaf flavonols, SPAD-relative leaf chlorophyll, $P_n$-net photosynthesis, $E$-Transpiration rate, $g_s$-stomatal conductance, $C_i$-internal CO2 concentration, WUE-water use efficiency, APX-ascorbate peroxidase, CAT-catalase, POX-peroxidase, SOD-superoxide dismutase, Prol-proline, Na-leaf $Na^+$, K-leaf $K^+$, Ca-leaf $Ca^{2+}$, Mg-leaf $Mg^{2+}$, Cl-leaf $Cl^-$, K/Na-leaf $K^+/Na^+$ ratio. GP_C: Goma Priyanka control, GP_S: Goma Priyanka sodic, J-37_C: CISH J-37 control, J-37_S: CISH J-37 sodic, J-42_C: CISH J-42 control, J-42_S: CISH J-42 sodic, KB_C: Konkan Bahadoli control, KB_S: Konkan Bahadoli sodic.

traits was largely consistent with their properties and functions. For instance, gas exchange traits, leaf $K^+$ and $K^+/Na^+$ ratio were clustered in tandem (lower left quadrant). The placement of leaf flavonols, proline, antioxidant enzymes, and harmful ions along PC2 indicated that osmotic and oxidative stress defenses were probably triggered in response to excess $Na^+$ and $Cl^-$. PC1 was able to differentiate between the control and sodicity treatments fairly clearly. The fact that PC2 could distinguish KB from other cultivars in a measurable way implied divergent responses to sodicity stress unique to KB. Interestingly, while cultivars responded quite differently under control treatment, J-37, J-42, and GP seemed to possess some shared reactions to deal with sodicity stress (Fig. 2).

## Ranking for sodicity tolerance

The membership function value (MFV) approach was used to analyze the sodicity tolerance of the tested cultivars. This approach provides a comprehensive assessment of cultivar's ability to withstand stress conditions by considering multiple traits

simultaneously. The MFV for each cultivar was calculated using the sodicity tolerance coefficients (STC) for all the measured traits (Table S4). It was found that none of the cultivars exhibited high tolerance, sensitivity, or extreme sensitivity to sodicity stress. With a mean MFV of 0.76, cultivar J-37 performed far better than others and was ranked as 'tolerant' to sodicity stress. The remaining cultivars were categorized as 'moderately tolerant' because their MFV values were all less than 0.50 (0.36 for J-42, 0.45 for GP, and 0.42 for KB).

## DISCUSSION

Soil dispersion, osmotic stress, restricted water and air movements, depletion of organic carbon, and decreased availability of essential nutrients are the major constraints to plant growth in sodic soils (*Singh et al., 2022a*). Salt-tolerant crops and cultivars play a crucial role in the sustainable management of sodic soils. There is barely any knowledge on how jamun plants respond to sodicity stress in terms of tree growth, gas exchange properties, accumulation of osmolytes and antioxidant enzymes, and Na$^+$ and Cl$^-$ accumulation. Moreover, to our knowledge, improved cultivars of jamun have not yet been evaluated comparatively under sodic conditions. Our study aimed to address these gaps by evaluating the effects of sodicity stress on growth, physiological relations, and ion uptake in jamun cultivars J-37, J-42, KB, and GP in order to delineate the mechanisms underpinning sodicity tolerance and to identify cultivars suitable for sodic soils.

The observed differences in vegetative growth under control conditions were likely due to diverse genetic backgrounds and innate variations in growth habit of the studied cultivars (*Lawande et al., 2014*; *Singh et al., 2018b*). Spreading growth habit in GP compared to semi-spreading in KB and upright in both J-37 and J-42 (*Protection of Plant Varieties and Farmer's Rights Authority, 2014*) implied a more vigorous growth in GP trees because spreading cultivars are frequently taller and wider, and tend to have larger trunk diameters than those with an upright growth (*Tworkoski & Miller, 2007*). Sodicity-induced reductions in TCSA, CV, and LA varied significantly among the cultivars. Harmful effects of sodicity stress vary remarkably among fruit varieties (*Singh et al., 2023a*; *Zhang et al., 2016*). Reduced soil osmotic potential in sodic soils causes trees to expend more energy for osmotic adjustment, limiting vegetative growth. Tree growth is further hindered by limited water and oxygen flows to the root surface (*Marino et al., 2019*), and excessive levels of Na$^+$ and Cl$^-$ (*Bhagwat & Kalbhor, 2023*). Our findings revealed significant increases in pH and ESP with soil depth. The alkaline pH in sodic soils decreases the availability of essential nutrients. Similarly, increase in ESP has negative impacts on soil aggregate stability, organic carbon content, nutrient recycling, and water availability (*Bhardwaj et al., 2019*). In sodic soils, organic carbon content and nutrient availability are inversely related to pH and ESP, and positively to CEC (*Jat et al., 2022*; *Meena et al., 2024*). Comparatively larger reductions in TCSA (>30%) indicated greater sensitivity of cultivars J-42 and KB to sodicity stress (*Kchaou et al., 2010*). Sodicity hampered canopy volume significantly in all cultivars except J-37, suggesting that J-37 trees have some adaptive mechanisms to cope with the excess salt (*Kumar et al., 2022*). Salt stress can limit leaf growth, potentially decreasing photosynthesis (*Negrão, Schmöckel & Tester, 2017*). However, studied jamun

cultivars showed only modest (<10.0%) reductions in leaf area under sodic conditions. Maintaining leaf area is a crucial adaptation under salt stress; it increases $CO_2$ transport to chloroplasts (*Zahra et al., 2022*) and reduces the risk of carbon starvation (*Simpson et al., 2014*).

Sodicity stress caused varying extents of decrease in relative chlorophyll (SPAD) except in cultivar J-37 in which SPAD values were non-significantly different between control and sodic treatments. Leaf SPAD in NaCl (100 or 200 mM) stressed olive cultivars was unaffected even 180 days after salt treatment (*Regni et al., 2019*). Salt stress negatively affects photosynthetic pigments, including chlorophyll; excess $Na^+$ and $Cl^-$ ions damage the pigment-protein complexes, ROS oxidize the chlorophyll pigments, and the enzyme chlorophyllase breaks down the chlorophyll molecule (*Behdad, Mohsenzadeh & Azizi, 2021*). The declines in $P_n$ under sodicity stress in jamun cultivars might be due to impaired leaf water relations (*Liu et al., 2020*) and increased accumulation of $Na^+$ and $Cl^-$ions (*Lu et al., 2022*). The downregulation of genes encoding essential photosynthetic enzymes as well as genes associated with photosystem structures and light-harvesting complexes may have contributed to such decreases (*Lin et al., 2018*). Cultivar GP, which had the highest $P_n$ under control treatment, showed a noticeably large decline (37.16%) in $P_n$ under sodicity stress (*Mousavi et al., 2019*). Sodicity-induced reductions in $P_n$ were comparatively less severe in both J-37 and J-42, probably due to upregulation of genes associated with carbon metabolism (*Hussain et al., 2012*). Sodicity stress also suppressed $E$, $g_s$ and $C_i$ to varying degrees in the tested cultivars. Reduced transpiration and increased accumulation of osmolytes are the key strategies to deal with salt-induced osmotic stress (*Yang et al., 2020*). Abscisic acid rapidly accumulates in salt-stressed plants to arrest the transpirational water loss *via* stomatal inhibition. Even though it is crucial for regulating ion uptake, reduced transpiration may inhibit plant growth because it is linked to the normal rates of photosynthesis (*Negrão, Schmöckel & Tester, 2017*). The salt-stressed plants improve their leaf water balance by decreasing the stomatal conductance and transpiration; this reduces the loss of water and improves WUE. Significant increases in WUE in cultivars J-37 and KB seemed to be due to reduced rates of $E$ and better maintenance of $P_n$ (*Upadhyay et al., 2018*).

It is speculated that increased anthocyanin and flavonoid accumulation protects the salt-stressed plants from oxidative stress, preserving their photosynthetic efficiency and growth. In particular, flavonoids are thought to hinder the formation of ROS. In transgenic plants, overexpression of genes involved in their synthesis boosted the accumulation of these compounds and improved the resistance of transgenics to oxidative and salt stresses, largely on account of better physiological activities (*Li et al., 2024*). In our study, cultivars KB and J-42 exhibited the largest (80.0%) and smallest (52.18%) upticks in Anth, respectively, under sodic conditions. The increases in Flav brought on by sodicity were relatively lower in cultivars J-37 (31.75%) and J-42 (29.23%), moderate in GP (40.30%) and substantial in KB (70.91%). Genetic variation for leaf phenolics, which function as compatible solutes to alleviate oxidative stress, is known in other crops (*Sarker & Oba, 2019*). However, Anth and Flav levels may not always increase linearly with increase in salt stress: Anth and Flav indices remained unaltered or even declined in salt-treated lettuce

cultivars (*Adhikari et al., 2021*), and their levels did not change in salt-stressed chilli and bell pepper plants (*Reimer et al., 2022*). In contrast, salt-stressed citrus rootstocks showed variable levels of surge in leaf phenolics; the largest increase was in the salt-sensitive Mexican lime and the lowest in the salt-tolerant Eingedi pummelo (*Hussain et al., 2012*). Increased accumulation of flavonoids under saline conditions hampered biomass production in certain lettuce cultivars, probably by impeding chlorophyll excitation and photosynthesis (*Adhikari et al., 2021*). In our study, cultivars KB and, to a lesser extent, GP which showed relatively larger increases in Anth and Flav also displayed comparatively greater declines in $P_n$, TCSA, and CV.

While ROS can function in activating the salt-stress responses at low concentrations, their higher levels can damage the vital cell components and biomolecules. Obviously, certain detoxification systems that mediate ROS scavenging are required to keep ROS levels below a threshold (*Yang & Guo, 2018*). Salt-stressed plants cope with ROS-induced oxidative stress by activating a variety of enzymatic (*e.g.*, SOD), and non-enzymatic (*e.g.*, flavonoids) antioxidants. SOD is usually the first line of defense; it dismutases superoxide anion ($O_2^-$) into $H_2O_2$ and $O_2$. APX and CAT then decompose $H_2O_2$ into $O_2$ and $H_2O$; APX frequently gets active at lower while CAT at higher concentrations of $H_2O_2$ (*Sarker & Oba, 2019*). Therefore, keeping ROS below their harmful levels requires the synergy between SOD and APX/CAT activities rather than merely their absolute levels. In our study, only J-37 and GP trees showed remarkable spikes in both SOD and CAT activities under sodic conditions, likely due to increased expression of genes linked to SOD and CAT functions (*Rasool et al., 2013*). Synergistic effects of SOD and CAT enhanced the tolerance of pear and peach cultivars to sodicity stress (*Singh et al., 2023a*). With the exception of APX activity in KB, cultivars J-42 and KB did not display notable rises in antioxidant enzymes when subjected to sodicity stress. This was little surprising because they had similar or even higher levels of antioxidant enzymes under control treatment in comparison to J-37 and GP. The damage to enzyme activity centers and/or suppression of enzyme expression due to high soil pH (*Yang et al., 2022*), and persistent ROS formation under long-term salt stress that impairs the antioxidant defense system (*Danaeifar et al., 2024*) may account for lower antioxidant enzyme activities in J-42 and KB trees. Different gene families in various cell organelles encode distinct antioxidant enzymes, and the expression of genes producing antioxidant enzymes varies with crops and genotypes (*Hasanuzzaman et al., 2021*). This can explain the cultivar-specific differences in antioxidant activity in the current study. Proline accumulation can improve plant salt tolerance by protecting Rubisco activity and mitochondrial electron transport chain, increasing water and nutrient uptake, boosting the antioxidant enzymes, reducing $Na^+$ and $Cl^-$ uptake, and increasing $K^+$ absorption (*El Moukhtari et al., 2020*). Under sodic conditions, while J-37 exhibited over three-fold increase, other cultivars showed only 1.5–2 times higher leaf proline than respective controls. Genes involved in proline biosynthesis were likely better activated while those underlying proline degradation were not upregulated by sodicity stress in cultivar J-37, whereas the reverse was true for other cultivars (*Hosseinifard et al., 2022*). This could have enhanced proline metabolism, ensuring higher proline levels in J-37 to counteract the effects of sodicity stress

(*Forlani, Bertazzini & Cagnano, 2019*). An alternative explanation may be that sodicity stress inhibited proline synthesis in J-42, KB and GP (*Yang et al., 2022*), compelling them to use other osmolytes for reducing the leaf osmotic potential (*Singh et al., 2023a*).

Sodicity stress significantly increased leaf $Na^+$ in all the cultivars; KB (170.99%) and J-37 (86.36%) exhibited the largest and smallest upticks relative to controls. Similarly, while J-37 showed the lowest rise (33.51%), cultivars J-42, GP, and KB demonstrated remarkably greater increases (>100%) in leaf $Cl^-$ under sodic conditions. Despite the widespread notion that rootstocks shield the shoots and leaves from salt injury, scion cultivars often have a significant role in regulating the ion uptake (*Simpson et al., 2014*; *Sivritepe et al., 2010*). While cultivars J-37 and GP were adept at limiting $Na^+$ absorption, J-37 was highly efficient in restricting $Cl^-$ uptake under sodic conditions. A plausible explanation for the lower $Na^+$ and $Cl^-$ build-up in J-37 leaves under sodic conditions is that trunk and canopy growth of J-37 were little impacted under sodic conditions, likely causing the absorbed $Na^+$ and $Cl^-$ ions to be redistributed throughout a larger tree biomass (*Zhang, 2014*). Apart from J-37 and KB, in which leaf $Na^+$ and $Cl^-$ levels were quite similar in control treatment, leaf $Na^+$ levels were mostly higher than $Cl^-$. This was likely because $Na^+$ transport is mostly unidirectional and there is little recirculation from shoots to roots (*Tester & Davenport, 2003*). Comparably, phloem recirculation seems to restrict $Cl^-$ accumulation, at least partly, in aerial organs (*Godfrey et al., 2019*). Sodicity-induced declines in leaf $K^+$ were negligible (<10.0%), except in GP (16.17%). Maintaining adequate $K^+$ and, consequently, a higher cytosolic $K^+/Na^+$ ratio may aid salt-stressed plants to cope with excess $Na^+$ (*Abid et al., 2020*). Sufficient $K^+$ levels may also enhance $Na^+$ sequestration into vacuoles (*Zarei et al., 2016*). Despite modest drops in leaf $K^+$, higher uptake of $Na^+$ under sodicity stress caused a decrease in the leaf $K^+/Na^+$ ratio in all the cultivars. Nonetheless, cultivar J-37 was least impacted. While sodicity stress raised $Ca^{2+}$ in J-37 leaves, it had no discernible effect on leaf $Ca^{2+}$ in J-42, KB, and GP. The decreases in leaf $Mg^{2+}$ in response to sodicity were pronounced only in J-42 (27.11%). Many crops preferentially accumulate or maintain $Ca^{2+}$ and $Mg^{2+}$ levels when exposed to salt (*Liu et al., 2020*). While $Ca^{2+}$ and $Mg^{2+}$ both boost osmotic adjustment (*Mahouachi, 2018*), $Ca^{2+}$ presumably also improves cell membrane stability (*Cimato et al., 2010*), and increases the selective uptake of $K^+$ over $Na^+$ (*Gengmao et al., 2015*).

We observed significant positive correlations among TCSA, CV, and LA; presumably because trees with bigger trunk diameters can translocate more water and nutrients for the enhanced growth of leaves and canopies (*Smith, 2008*). Significant positive correlations of TCSA and CV with $P_n$ were likely because more leaves and a bigger canopy in trees with larger trunk diameters increase the surface area for photosynthesis (*de Mattos et al., 2020*). Larger leaves probably contribute more to photosynthesis because of their larger surface area for light interception, higher density of mesophyll cells, and increased gas exchange capacity. The negative correlations of TCSA, CV and LA with phenolics and antioxidant enzymes highlight trade-off between tree growth and salt-induced secondary metabolism (*Yang et al., 2024*). The correlations between gas exchange attributes on the one hand and leaf phenolics, antioxidant enzymes, and proline on the other were also mostly significantly negative. There exists a negative correlation between the levels of leaf phenolics and net

photosynthetic rate in many crops (*Kostidis & Karabourniotis, 2024*). Similarly, inverse relationships of leaf proline with growth and photosynthetic traits suggested that proline levels were probably rather low to have any meaningful effect on osmotic adjustment (*Ismail et al., 2016*; *Yang et al., 2024*). Apart from decrease in photosynthesis, the energy costs associated with various salt tolerance mechanisms can negatively impact plant growth under salt stress (*Munns & Gilliham, 2015*). Energy expenditure for proline accumulation occurs at the cost of plant growth (*Mansour, 2020*). Interestingly, both TCSA, CV, and LA exhibited stronger negative correlations with leaf $Cl^-$ than with leaf $Na^+$, implying that increased leaf $Cl^-$ levels may more adversely impact jamun tree growth under sodic conditions. The mechanisms to deal with excess $Na^+$ and $Cl^-$ including their restricted uptake and sequestration in vacuoles require additional energy, negatively affecting plant growth (*Litalien & Zeeb, 2020*). While both excess $Na^+$ and $Cl^-$ may inhibit tree growth (*Simpson et al., 2014*), increased $Cl^-$ may be more harmful (*Liu et al., 2020*) because $Cl^-$ ions are not absorbed in soil and are easily absorbed by plants (*Stavi, Thevs & Priori, 2021*). $Cl^-$ gets transported to the leaves *via* sap flow, increasing their osmotic potential and decreasing water availability for metabolism (*Stavi, Thevs & Priori, 2021*). Use of organo-mineral fertilizers and organic inputs in addition to avoiding fertilizers that contain chlorine may be advantageous under such conditions (*Mohanavelu, Naganna & Al-Ansari, 2021*). Relatively small reductions in leaf $K^+$ under sodic conditions might also have partially offset the negative effects of $Na^+$ on tree growth by maintaining a higher $K^+$/$Na^+$ ratio (*Abid et al., 2020*). Adequate $K^+$ levels improve osmotic adjustment, maintain the cell turgor, inhibit the excessive production of ROS, and aid in the induction of programmed cell death in salt-stressed plants (*Hussain et al., 2021*). Calcium and magnesium play structural and functional roles in the leaves, and their increased accumulation may support the metabolic demands of a larger canopy. Both $K^+$ and $Ca^{2+}$ play diverse physiological functions in plants, including the maintenance of cell osmotic pressure which may explain their strong positive correlations with leaf area. A significant correlation between leaf $Mg^{2+}$ and photosynthetic rate can be explained by the fact that $Mg^{2+}$ is required for chlorophyll synthesis, and plays an important role in photosynthesis and related processes (*Liu et al., 2020*).

Multivariate approaches including PCA are typically better suited for detecting important patterns in data involving multiple (multicollinear) variables (*Julkowska et al., 2019*). Furthermore, graphical representation of PCA loadings makes it easier to distinguish between the shared and contrasting growth and physiological responses to salt stress (*Singh et al., 2022b*). In our study, PCA efficiently compressed dimensionality and detected cultivar- and sodicity-specific effects in data, revealing some interesting insights into how jamun cultivars respond to sodicity stress. While PC1 efficiently distinguished between control and sodicity treatments, PC2 differentiated KB from the other cultivars. Notably, PCA revealed some shared reactions among cultivars J-37, J-42, and GP to cope with elevated soil sodicity (*Abid et al., 2020*; *Singh et al., 2022b*). The application of membership function analysis (MFA) is increasingly gaining traction for comprehensive evaluation of salt tolerance in crop plants. Since MFA ranks salt tolerance based on all the studied parameters, it is apparently more robust than indices such as salt tolerance index

or stress susceptibility index which are based on a single attribute. MFA also considers the fact that salt tolerance is rarely a binary response because plants exhibit varying degrees of tolerance rather than a distinct 'tolerant' or 'sensitive' response (*Gholizadeh et al., 2022*; *Tian et al., 2024*). Our findings revealed contrasting variation for tolerance to sodicity stress. Cultivar J-37 exhibited the highest mean MFV of 0.76, and was ranked as 'tolerant' to sodicity stress. Of the remaining cultivars, Goma Priyanka also demonstrated relatively better tolerance with the mean MFV of 0.45. In contrast, the mean MFV values were considerably lower for Konkan Bahadoli (0.42) and CISH J-42 (0.36) indicating their greater sensitivity to sodicity stress, particularly in comparison of cultivar CISH J-37. Maintenance of photosynthetic rate, increased leaf proline levels, greater and synergistic activities of SOD and CAT, and a higher $K^+/Na^+$ ratio in leaves seemed to be the major physiological underpinnings for tolerance to sodicity stress in cultivar J-37 (*Pathania et al., 2023*).

## CONCLUSIONS

Contrasting responses to sodicity stress of jamun cultivars has significant implications for researchers, farmers, and policymakers. While sodicity stress had a significant negative impact on cultivars CISH J-42, Konkan Bahadoli, and Goma Priyanka, CISH J-37 was least affected. Such divergent reactions indicate genetic variability for salt tolerance, which can be harnessed in breeding programs to develop varieties with improved sodicity tolerance. While cultivars such as CISH J-37 may be suitable for sodic regions, others may be preferred in areas with minimal sodicity stress. Agricultural and extension agencies can focus on promoting sodicity tolerant jamun cultivars for enhanced food and biomass production, and ecological sustainability in sodic soil areas. Our results open the avenues for further research to delineate the physiological and genetic mechanisms underpinning sodicity tolerance in jamun. Specifically, the ion channels and transporters involved in $Na^+$, $Cl^-$ and $K^+$ absorption and translocation need to be identified. Evaluating the long-term effects of sodicity stress on fruit yield and quality may be another important area of research.

## ABBREVIATIONS

| | |
|---|---|
| **ANOVA** | Analysis of Variance |
| **Anth** | anthocyanin |
| **APX** | Ascorbate peroxidase |
| **CAT** | Catalase |
| **CEC** | cation exchange capacity |
| $C_i$ | Internal $CO_2$ concentration |
| **CV** | canopy volume |
| *E* | Transpiration |
| $EC_e$ | Soil saturation extract electrical conductivity |
| **ESP** | exchangeable sodium percentage |
| **Flav** | Flavonols |
| **GP** | Goma Priyanka |

| $g_s$ | Stomatal conductance |
|---|---|
| **J-37** | CISH J-37 |
| **J-42** | CISH J-42 |
| **KB** | Konkan Bahadoli |
| **OC** | organic carbon |
| **PCA** | Principal Component Analysis |
| $pH_s$ | Saturated soil paste pH |
| $P_n$ | Net photosynthesis |
| **POX** | Peroxidase |
| **ROS** | Reactive oxygen species |
| **SOD** | Superoxide dismutase |
| **SPAD** | Soil Plant Analysis Development |
| **TCSA** | Trunk cross sectional area |
| **WUE** | Water use efficiency |

## ACKNOWLEDGEMENTS

We kindly acknowledge late Dr. Sanjay Singh for providing the experimental plants. Dheeraj Kumar is thanked for his technical support.

### Funding

This study was supported by ICAR-CSSRI, Karnal, Haryana, India. The funders had no role in study design, data collection and analysis, decision to publish, or preparation of the manuscript.

### Grant Disclosures

The following grant information was disclosed by the authors:
ICAR-CSSRI, Karnal, Haryana, India.

### Competing Interests

Anshuman Singh is an Academic Editor for PeerJ Life and Environment.

### Author Contributions

- Anshuman Singh conceived and designed the experiments, performed the experiments, analyzed the data, prepared figures and/or tables, authored or reviewed drafts of the article, and approved the final draft.
- Ashwani Kumar performed the experiments, authored or reviewed drafts of the article, and approved the final draft.
- Jai Prakash analyzed the data, prepared figures and/or tables, authored or reviewed drafts of the article, and approved the final draft.
- Daya Shankar Mishra analyzed the data, authored or reviewed drafts of the article, and approved the final draft.

## Data Availability

The raw data are available in the Supplemental File.

## Supplemental Information

Supplemental information for this article can be found online at http://dx.doi.org/10.7717/peerj.19132#supplemental-information.

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
