# Peer review of "Physiological traits underlying sodicity tolerance in Jamun (Syzygium cumini L. Skeels) cultivars"

_PeerJ, doi:10.7717/peerj.19132_

## Round 0.1 · original submission · Major Revisions

Dear Authors

The manuscript cannot be accepted for publication in its current form. It needs substantial revision to meet the journal's standards. The authors are invited to revise the paper, considering all the suggestions made by all the reviewers, including those who rejected the manuscript. Please note that requested changes are required for publication.

With Thanks

Reviewer 1 ·

Basic reporting

The introduction lacks a cohesive structure. There is a sudden shift from discussing soil salinity problems to jamun’s bioactive compounds, which seems disconnected from the main research focus on sodicity stress and plant physiology. Make sure each paragraph logically connects to the next, building towards a clear argument or justification for the study.
The choice of jamun as the study subject is stated but not fully justified in terms of its relevance to solving the broader problem of sodicity in agriculture. The introduction mentions its nutraceutical potential but does not clearly explain why jamun is particularly interesting or valuable for sodicity tolerance studies. A more compelling case could be made about why jamun’s salt tolerance is of special interest in the context of both crop productivity and environmental sustainability in sodic soils.
Some sections repeat information unnecessarily. For instance, salt tolerance mechanisms are discussed in multiple places, and the emphasis on gypsum application in sodic soils is reiterated several times without adding new insights.
In Materials and Methods section, provide details about how many samples were taken, how they were composited, and whether they were mixed before analysis. More specific information on the sampling distances relative to trees and replication numbers would strengthen the clarity of the experimental design.
It is mentioned that the site is prone to waterlogging, yet no mitigation strategies (such as drainage systems or raised beds) are described to control this confounding factor, which could significantly affect tree growth and obscure results related to sodicity stress.
Please provide the details of methods used to measure antioxidant enzymes and proline content along with details of specific equipment used and sample preparation. Why enzyme activity was measured in units/ g fresh weight rather than units/mg protein?
The description of the membership function analysis is highly mathematical but not intuitive for most readers. It would benefit from a clearer explanation of how this analysis helps assess sodicity tolerance compared to more traditional metrics like tolerance indices or stress susceptibility indices. Furthermore, the formula could be better contextualized to make its application in plant research clearer, particularly how it translates into practical recommendations for crop improvement.
The study spans over two years (2018–2020), but no discussion of potential seasonal variation in growth and physiological responses is included. These seasonal factors could introduce significant variability into the results and should be addressed, especially when comparing growth responses measured in different seasons.
In results section, the MFV approach for ranking sodicity tolerance is an interesting addition, but it is not well integrated into the narrative. The conclusion that "none of the cultivars exhibited high tolerance or extreme sensitivity" is underdeveloped and appears abrupt, given the range of responses observed.
Although the discussion touches on various physiological responses to sodicity stress (e.g., proline, flavonoids, SPAD values, ion uptake), the physiological mechanisms and pathways underlying these responses are not deeply examined. For instance, the increase in proline levels is mentioned but not linked to detailed cellular processes that might explain why some cultivars accumulate more proline. The roles of these physiological adaptations in osmotic balance, cell stability, or ROS detoxification could be elaborated further.
The discussion lacks emphasis on what is novel or groundbreaking about the current study’s findings. Repeated references to prior studies, while necessary for context, overshadow the unique contributions of this study. It would be beneficial to clearly distinguish where this study fills gaps in the literature.
The mention of "strong negative relationships" between growth attributes and leaf Cl- is under-explained. It’s unclear whether this is a novel finding, and the implications of these correlations are not thoroughly discussed. More detailed discussion on how these correlations impact practical cultivation strategies should be included.
The conclusion does not adequately address the practical implications for growers or policymakers. Practical recommendations for selecting jamun cultivars for sodic soils or strategies for improving tolerance could be included, providing more direct applications for the study’s findings.
All Figures and Tables are well presented.
Specific Comments:
114-115: All the cultivars were grafted on uniform rootstocks raised using seeds from a single mother tree. There is no information regarding the age of the mother tree or whether the rootstocks were uniformly grown. The variation in rootstock can significantly influence plant growth, and it is essential to describe the uniformity of the rootstocks and scions used.
119-120: “Young plants were properly trained to develop a strong canopy framework; only 3-5 well spaced scaffold branches were allowed to grow above 60 cm from the ground surface.” Clarify it by adding details.
120-122: “Subsequent to planting, each tree was fertilized using 125 g N, 50 g P2O5 and 50 g K2O in two equal splits (Singh et al., 2010). Provide details of splits.
124-126 “The groundwater used for irrigation had an electrical conductivity of 0.68 dS/m, pH of 8.04, Na+ of 2.43, K+ of 0.16, Ca2+ + Mg2+ of 4.14, ClF of 0.89 and HCO3 of 4.22 (all ions in meq/l).” Could you please confirm whether the groundwater used in the experiment was the sole source of sodium ions, or if the soil itself contained sodium as well? Additionally, was the soil naturally sodic, or was it artificially made sodic for the purpose of the experiment?
137-138 “The corresponding ECe values were 0.58, 0.73 and 0.92 dS/m in control, and 0.66, 0.95 and 1.14 dS/m in sodic treatment” Please provide the Exchangeable Sodium Percentage (ESP), Cation Exchange Capacity (CEC), and Organic Matter content for both the control and sodic treatment. This data is essential to critically assess the validity of the claims made by the authors and to understand how the soil’s condition could affect plant performance.
140-141: “Six trees (n= 6) of each jamun cultivar within both control and sodicity treatments were tagged for recording the observations.” Could you please clarify whether this refers to 6 biological replicates per treatment?
188-190 “A two-way Analysis of Variance (ANOVA) was used to examine the independent and interaction effects of cultivar and sodicity on different traits. The means were compared by Tukey test (p < 0.05). Could you clarify how the use of a two-way ANOVA is justified based on your experimental design?

Experimental design

Critical information regarding rootstock uniformity, soil properties including ESP (exchangeable sodium percentage), and method of sodium treatment must be clearly provided to assess the robustness of the study and the reliability of its conclusions. How authors maintained the control conditions infect the irrigation water also contained sodium content. Please provide the details of statistical softwares and versions used for ANOVA and correlation and PCA analysis.

Validity of the findings

The results are promising, subject to the authors providing clarification on the queries as mentioned in the basic reporting.

Reviewer 2 ·

Basic reporting

The work lacks of novelty. Most of the work were routine analyses.

Experimental design

OK.

Validity of the findings

The novelty of this work was limited.

Additional comments

peerj-reviewing-106425-v0
The report was a routine work. There were some similar reports with similar approaches, though with different agricultural products and/or cultivars. The in-depth reason of cultivar J-37 showed different results needs further investigation.
The internal relationship among different parameters needs more elaboration. Some of them should be closely related to each other. Thus, the presentation of the results and corresponding discussion should be more on elaborating the relationship.

Reviewer 3 ·

Basic reporting

Thank you for providing me the opportunity to review the manuscript ‘Physiological traits
underlying sodicity tolerance in Jamun (Syzygium cumini L. Skeels). In my opinion, this is an important contribution to this field and deserves publication. The paper is well organized and mostly well authored. However, I have observed some deficiencies which should be addressed before the manuscript is accepted for publication.

Experimental design

Good and very much explained.

Validity of the findings

Very good

Additional comments

Added as separate file

Annotated reviews are not available for download in order to protect the identity of reviewers who chose to remain anonymous.

---

## Round 0.2 · Minor Revisions

Dear Authors
The manuscript still needs a minor revision before publication. The authors are invited to revise the paper considering all the suggestions made by the reviewers. Please note that the requested changes are required for publication.
Best Regards

Reviewer 1 ·

Basic reporting

I appreciate that the authors have satisfactorily addressed most of the previous comments. However, the manuscript still requires minor improvements.
• There are a few typographical errors, such as spacing issues. For example, refer to Line Nos. 58 and 94.
• I recommend carefully reviewing the entire manuscript to identify and correct similar errors.
• Please correct this, Line No. 98-101: “This is mainly because earlier studies evaluated the suitability of locally-adapted non-descript jamun genotypes vis-à-vis other fruit and forestry trees 100 for sodic soils purely in terms of plant growth, without considering the sodicity-induced physiological changes (Gill and Abrol, 1991; Singh et al., 1997; Singh et al., 2008).” change to “This is primarily because earlier studies assessed the suitability of locally adapted, non-descript jamun genotypes in comparison to other fruit and forestry trees for sodic soils solely based on plant growth, without considering sodicity-induced physiological changes (Gill and Abrol, 1991; Singh et al., 1997; Singh et al., 2008).” The authors have cited outdated references for this information. Please replace these with more recent and relevant studies from peer-reviewed literature.
• Line No. 90-92 “Plants typically negate such deleterious effects by excluding Na+ and Cl− ions, maintaining leaf K+ levels, accumulating the compatible osmolytes like proline, and upregulating the anti-oxidant enzymes.” requires appropriate references to support the claims made. Please include relevant citations from peer-reviewed literature to substantiate these points.
• Please clarify the methodology and provide details on how the cation exchange capacity (CEC) and exchangeable sodium percentage (ESP) were calculated or measured based on sodium concentration. Ensure to establishes a clear relationship between the measured sodium concentration and the derived parameters (CEC and ESP).

Experimental design

Experimental design seems appropriate.

Validity of the findings

Findings seems valid.

---

## Round 0.3 · Minor Revisions

Dear Authors
The manuscript still needs a minor revision before publication. The authors are invited to revise the paper considering all the suggestions made by the reviewers. Please note that the requested changes are required for publication.
With Thanks

Reviewer 1 ·

Basic reporting

The authors have addressed most of the queries raised during the previous round of revision, and the manuscript has shown significant improvement. However, a few minor adjustments are still required for better clarity and technical accuracy:
Material and Methods Section:
Include all equations in proper equation formatting using equation.
Line 157-158: The equation CEC = (Na concentration of extract in me/L × 10)/weight of soil sample (g) needs further clarification. Provide detailed explanations of the symbols and terms used in this equation for proper understanding.
Line 180-181: The description of leaf phenolics measurement using the Dualex 4 Scientific leaf-clip sensor (FORCE-A, Orsay Cedex, France) is insufficient. Add a brief explanation of the method, including its working principle and significance.
Add volumes used for SOD activity in the reaction mixture.
Line 258-259: Clarify the traits inversely related to sodicity tolerance (leaf Na⁺ and Cl⁻) measured using MFV2. Additionally, elaborate on the terms Xi and X¯ (mean values?) in the context of the membership function. Clearly explain the concept of membership function.

Experimental design

Experimental design is good.

Validity of the findings

Findings seems promising.

Reviewer 3 ·

Basic reporting

The author have revised the manuscript thoroughly and did satisfactorily well. Now the MS seems to be in order to be publish in high impacted journal like PeerJ.

Experimental design

Adequate and understandable.

Validity of the findings

Finding are also interesting and valid

Additional comments

The manuscript is thoroughly revised and is in order to be publish. The recommendation is accepted for the present MS.

---

## Round 0.4 · Minor Revisions

Dear Authors

The manuscript still needs a minor revision before publication. The authors are invited to revise the paper considering all the suggestions below. Please note that the requested changes are required for publication.

"The paper describes the response to "control" and "sodic" treatments, but the methods do not provide information on what the sodic treatment was or how it was applied.

The "background" section of the abstract is insufficient to explain the motivation of the study and should be expanded. The authors can look to other PeerJ Plant Bio articles for examples.

The first paragraph of the introduction alternates between discussing salt stress and sodicity without explaining the connection between them. This can confuse non-expert readers; please explain the relationship.

With Thanks

Reviewer 1 ·

Basic reporting

The authors have addressed most of the recommended suggestions in the previous round of revision, and the revised manuscript has been significantly improved. The article is now suitable for publication in PeerJ.

Experimental design

Experimental design seems promising.

Validity of the findings

results seems valid.

---

## Round 0.5 · accepted · Accept

Dear Authors,

I am pleased to inform you that the manuscript has improved after the last revision and can be accepted for publication.

Congratulations on accepting your manuscript and thank you for your interest in submitting your work to PeerJ.

With Thanks